# Diminished Short-Term Efficacy of Reduced-Dose Induction BCG in the Treatment of Non-Muscle Invasive Bladder Cancer

**DOI:** 10.3390/cancers15143746

**Published:** 2023-07-24

**Authors:** David A. Ostrowski, Raju R. Chelluri, Matthew Herzig, Leilei Xia, Brian D. Cortese, Daniel S. Roberson, Thomas J. Guzzo, Daniel J. Lee, S. Bruce Malkowicz

**Affiliations:** 1Division of Urology, Department of Surgery, University of Pennsylvania Health System, Philadelphia, PA 19104, USA; david.ostrowski@pennmedicine.upenn.edu (D.A.O.);; 2Division of Urology and Urologic Oncology, Fox Chase Cancer Center, Philadelphia, PA 19111, USA; 3Department of Medicine, Beth Israel Deaconess Medical Center, Boston, MA 02115, USA

**Keywords:** BCG shortage, reduced dose, bacillus Calmette-Guérin, non-muscle invasive bladder cancer

## Abstract

**Simple Summary:**

Intravesical Bacillus Calmette-Guérin (BCG) instillation is an important treatment for non-muscle invasive bladder cancer. Recent BCG shortages have forced urologists to try alternative intravesical BCG regimens to maximize a limited supply. The goal of our study was to compare the efficacy of reduced-dose induction BCG versus full-dose induction BCG in regards to bladder tumor recurrence rate. We hypothesized that patients receiving reduced-dose induction BCG would have a higher recurrence rate. In our single-center retrospective cohort of 139 patients meeting the inclusion criteria, 38.6% of patients who received reduced-dose induction BCG developed a recurrence within one year compared to 33.7% of propensity-matched patients who received full-dose induction BCG. We concluded that reduced-dose induction BCG was associated with a significantly greater risk of recurrence within one year than full-dose induction BCG therapy. These data suggested that reduced-dose induction BCG may not be equivalent or non-inferior to full-dose administration in the short term.

**Abstract:**

The ongoing Bacillus Calmette-Guérin (BCG) shortage has created challenges for the treatment of non-muscle invasive bladder cancer (NMIBCa). Our objective was to evaluate the efficacy of reduced-dose induction BCG (RD-iBCG) compared to full-dose induction BCG (FD-iBCG) regarding recurrence rates. We hypothesized that patients receiving RD-iBCG may recur at a higher rate compared to those who received FD-iBCG therapy. A retrospective review of all patients with NMIBCa treated with intravesical therapy at our institution between 2015–2020 was conducted. Inclusion criteria consisted of having a diagnosis of AUA intermediate or high-risk NMIBCa with an indication for a six-week induction course of FD or RD-BCG with at least 1 year of documented follow up. The data were censored at one year. Propensity score matching for age, sex, tumor pathology, and initial vs. recurrent disease was performed. The primary endpoint was bladder cancer recurrence, reported as recurrence-free survival. A total of 254 patients were reviewed for this study. Our final cohort was 139 patients after exclusion. Thirty-nine percent of patients had HGT1 disease. 38.6% of patients receiving RD-BCG developed a recurrence of bladder cancer within a one-year follow-up as compared to 33.7% of patients receiving FD therapy. After propensity matching, this value remained statistically significant (*p* = 0.03). In conclusion, RD-iBCG for NMIBCa is associated with a significantly greater risk of recurrence than full-dose induction therapy, suggesting that RD-iBCG may not be equivalent or non-inferior to full-dose administration in the short term.

## 1. Introduction

Bladder cancer is one of the ten most common cancers diagnosed globally and is a major source of morbidity and mortality as well as the cause of a significant economic burden [1,2]. Urothelial carcinoma is the most common histology found. Approximately 75% of diagnosed bladder cancers are reported as non-muscle-invasive bladder cancer (NMIBCa) after initial resection [3,4]. Identification of NMIBCa represents an important opportunity for intervention, as in cases of high-risk NMIBCa, up to 25% of patients may progress to muscle-invasive or metastatic disease with a resultant pronounced decrease in 5-year overall survival [5,6].

Maximal transurethral resection of bladder tumor (TURBT), which includes resection into muscularis propria and clearance of all endoscopically identified tumors, is the current first step in the standard of care treatment regimen for NMIBCa [7,8,9]. However, maximal TURBT alone is limited in its ability to fully treat NMIBCa with recurrence rates up to 70% and progression rates up to 40%, with increased recurrence and progression rates with a higher pathologic grade of bladder cancer, when TURBT is performed without subsequent adjuvant therapy [10,11]. In order to reduce rates of recurrence and progression for patients with intermediate- and high-risk NMIBCa, current clinical guidelines recommend adjuvant induction and maintenance of intravesical Bacillus Calmette-Guérin (BCG) [8].

First described by Morales et al. in the 1970s, intravesical instillation of BCG seems to cause local immunomodulation in the bladder by recruiting macrophages, NK cells, CD8 T cells, and CD4 T cells into the urothelium, resulting in an anti-tumor effect, although the exact mechanism is unclear [12,13]. Compared to TURBT alone, a six-week induction course of intravesical BCG has been demonstrated to improve recurrence-free and progression-free survival rates for patients with NMIBCa [14,15,16,17]. In addition, maintenance intravesical BCG following the induction course further reduces the risk of tumor recurrence for intermediate- and high-risk NMIBCa patients [18,19,20,21]. At each instillation of BCG therapy, the accepted standard BCG dose is what was initially described by Morales et al. based on animal studies and clinical trials to achieve the intravesical introduction of 10^6^–10^8^ colony forming units that were thought to be required to achieve an anti-tumor effect [12,22].

While BCG is an integral component in the treatment of intermediate- and high-risk NMIBCa, a global shortage of BCG has made the acquisition of adequate BCG challenging. BCG production has been dwindling for years due to the difficulty of production, limitations on production capacity, financial disincentives, and regulatory hurdles [23,24]. The global BCG supply then drastically decreased when Sanofi Aventis ceased production of the Connaught strain in 2017 [25]. While Merck increased production of the TICE strain, the months-long production time of each BCG batch has limited the ability of the available BCG supply to match the current demand in many parts of the world [26]. In this setting of BCG shortages, there is a need for alternative intravesical treatment regimens for patients with intermediate- and high-risk NMIBCa.

One potential option for adapting to a limited BCG supply is by stretching the available BCG resource by decreasing the dosage of BCG instilled at each treatment or decreasing the frequency of instillations, so the dosing and timing of intravesical BCG has been an ongoing area of investigation. Results from the Spanish Urology Association for Oncological Treatment (CUETO) group and the European Organization for Research and Treatment of Cancer (EORTC) 30926 have suggested equivalent or non-inferior efficacy of reduced-dose induction BCG compared to full-dose BCG in terms of five-year disease-free rates for induction or induction plus one year of maintenance [19,22,27]. In contrast, two recent meta-analyses reported conflicting conclusions regarding recurrence rates for reduced-dose induction BCG [28,29]. In addition, the Treatment of High-grade Non-muscle-invasive Bladder Carcinoma by Standard Number and Dose of BCG Instillations Versus Reduced Number and Standard Dose of BCG Instillations (NIMBUS) trial demonstrated inferiority of decreasing the frequency of standard-dose BCG instillations (induction therapy at 1, 2 and 6 weeks followed by 2 weeks of maintenance at 3, 6 and 12 months) compared to standard therapy timing (6 weeks of induction therapy followed by 3 weeks of maintenance at 3, 6 and 12 months) [30]. In the setting of mixed evidence, reduced-dose induction BCG has been proposed as an alternative to full-dose induction, if needed, due to local supply constraints.

Other alternative options include the use of intravesical chemotherapeutic agents to completely replace BCG. Intravesical mitomycin C is one such potential option. When compared to BCG, a large randomized controlled trial performed by the Southwest Oncology Group (SWOG) demonstrated inferior recurrence-free survival for mitomycin C among patients with Ta and T1 NMIBCa, although there were no differences in progression-free survival or overall survival [31]. However, a Cochrane review of multiple studies comparing intermediate- and high-risk NMIBCa patients, including multiple randomized controlled trials, demonstrated no difference in overall survival, progression-free survival, or recurrence-free survival between patients treated with intravesical BCG or intravesical mitomycin C as well as a potentially increased rate of serious adverse events for patients treated with BCG compared to patients treated with mitomycin C, although the authors noted significant heterogeneity between trials and varying follow-up intervals and treatment schedules [32]. In addition, from subgroup analyses of multiple trials, including maintenance regimens, a decreased risk of recurrence has been reported for patients receiving BCG maintenance compared to mitomycin C maintenance [33].

Intravesical gemcitabine is another chemotherapeutic option that has been explored in NMIBCa. Compared to mitomycin C, a potential benefit in recurrence-free survival has been reported for patients receiving intravesical gemcitabine [8]. When comparing gemcitabine with BCG, randomized trials have reported no difference in all-cause mortality [34] or progression-free survival [34,35]. In terms of recurrence rates, three randomized trials reported mixed results that were limited by patient heterogeneity and inconsistency of results, making clear conclusions about this medication’s efficacy in reducing recurrence difficult [33,34,35,36]. Other single agent intravesical regimens using thiotepa, doxorubicin or epirubicin have also been evaluated, with each of these agents demonstrating inferior recurrence-free survival and/or progression-free survival compared to intravesical BCG [37,38,39]. A dual intravesical chemotherapy regimen of sequential gemcitabine and docetaxel has shown promise as rescue therapy in a retrospective cohort of BCG-unresponsive patients [40] as well as a potential alternative adjuvant therapy in a small cohort of BCG-naïve patients with high-grade NMIBCa [41]. In addition, a recent single-center retrospective cohort study reported improved high-grade recurrence-free survival and less treatment discontinuation for patients with high-risk NMIBCa treated with sequential intravesical gemcitabine and docetaxel versus intravesical BCG [42]. While current high-quality evidence for sequential intravesical gemcitabine and docetaxel is limited, there is a currently enrolling large randomized controlled phase III trial, the BRIDGE trial, designed to compare intravesical BCG versus intravesical gemcitabine and docetaxel in patients with high-grade NMIBCa to better evaluate this dual chemotherapy regimen as a BCG alternative [43,44].

Given the ongoing BCG shortage, the long history of established efficacy of induction and maintenance of BCG in intermediate- and high-risk NMIBCa, and the limited and inconsistent data for intravesical chemotherapeutic regimens as alternatives for BCG, the American Urological Association (AUA) and the Bladder Cancer Advocacy Network (BCAN) released recommendations in 2020 to guide urologists faced with limited BCG supply [45]. They recommended intravesical chemotherapy, rather than intravesical BCG, as a first- and second-line adjuvant therapy following TURBT for patients with intermediate-risk NMIBCa. They recommended prioritization of full-strength induction BCG therapy for high-risk NMIBCa, high-grade T1, and carcinoma in situ (CIS) patients with dose reduction to 1/2–1/3 dose induction BCG if full-strength dosing is not available. For high-risk NMIBCa patients, maintenance intravesical BCG therapy for 1 year (instead of the classic 3-year schedule) was recommended if supply allowed. However, prioritization of induction BCG over maintenance BCG, if needed, is recommended. When BCG is not available, AUA/BCAN recommendations suggest consideration of intravesical chemotherapy for induction and possible maintenance for high-risk NMIBCa patients.

Urologists have had to make difficult resource allocation decisions based on relatively limited data and with limited recommendations from professional societies when facing a limited BCG supply. Beginning in 2019, during a local BCG supply shortage, our institution, closely reflecting what would later become the AUA/BCAN BCG supply shortage recommendations, began offering reduced-dose induction BCG without maintenance BCG to patients with AUA intermediate- or high-risk NMIBCa. In this study, we evaluate tumor response to reduced-dose induction BCG in the real-world setting of a pragmatic university practice facing BCG allocation limitations by comparing these cases to a prior, propensity-matched series of patients with intermediate- and high-risk NMIBCa who received full-dose induction BCG therapy. We hypothesized that patients receiving reduced-dose induction BCG may recur at a higher rate compared to those who received full-dose induction BCG.

## 2. Materials and Methods

### 2.1. Ethics Statement

This retrospective cohort study was reviewed and approved by the University of Pennsylvania Institutional Review Board (IRB). Given the retrospective and de-identified nature of this study, the University of Pennsylvania IRB approved a waiver of informed consent.

### 2.2. Study Design and Population

A retrospective cohort study was performed on a population of adult patients with intermediate- or high-risk NMIBCa treated with intravesical induction therapy at our institution between the years 2015 and 2020. Using our institution’s database, we identified patients with intermediate- or high-risk NMIBCa and retrospectively reviewed the electronic medical record (EMR) to determine whether the patients were eligible for this study cohort. Inclusion criteria consisted of having a diagnosis of AUA intermediate or high-risk NMIBCa following transurethral resection of bladder tumor (TURBT) and restaging TURBT (as clinically indicated) with an indication for a six-week induction course of full-dose or reduced-dose induction BCG with at least 365 days of documented follow-up. Exclusion criteria included no receipt of BCG, AUA low-risk NMIBCa, muscle invasive bladder cancer, receipt of therapy at an outside institution, partial/complete BCG induction prior to presentation to our institution, and therapies other than TURBT or BCG. Figure 1 illustrates the generation of the cohort. Data were censored at the one-year follow-up.

### 2.3. BCG Administration and Clinical Follow-Up

BCG dose administration was based on allotment availability; there was no standardization for a particular patient for what dose they received. Therefore, the reduced-dose cohort of patients received either a 1/3 dose, a 1/2 dose, or a dose between these two doses. All patients completed an induction course of BCG. No patients received maintenance BCG, as an institutional allocation decision was made to prioritize induction therapy in the setting of a limited BCG supply. During this time, patients were offered an alternative intravesical maintenance regimen of gemcitabine and docetaxel. Patients were not excluded based on whether they received alternative intravesical maintenance. All patients underwent follow-up surveillance cystoscopy at intervals determined by their urologist per guidelines. Recurrences were defined as the identification of a new bladder tumor on cystoscopic surveillance within one year after completion of their induction BCG regimen. Data were captured upon the patient’s first presentation to our institution’s urology department. Review of follow-up was undertaken for all patients in the cohort until the last documented visit with urology in the EMR.

### 2.4. Propensity Matching

Propensity score matching for age, sex, tumor pathology, and initial vs. recurrent disease was performed in an attempt to decrease bias in our retrospective cohort using the method utilized previously by our group, in keeping with literature standards [46,47,48]. The matching procedure was performed using R statistical software version 4.1.0 (R foundation; Vienna, Austria) and the R Matching package [48]. In the matching procedure, logistic regression was used to model the probability of each patient being treated with reduced-dose induction BCG using the covariates of age, sex, tumor pathology, and initial vs. recurrent disease. For each patient, the propensity score, which was defined as the probability of receiving reduced-dose BCG induction, was computed using the logistic regression model. Then, patients who received reduced-dose induction BCG were matched 1:1 with patients who received full-dose induction BCG therapy based on the nearest neighbor match of the propensity score.

### 2.5. Statistical Analysis

The primary endpoint was bladder cancer recurrence within one year on surveillance cystoscopy, which was reported as recurrence-free survival. Comparisons of covariates were performed via the two-sample Student’s *t*-test (normality assessed by evaluating the histogram distribution shapes of the data) for continuous variables and the chi-square test for categorical variables. Cox proportional hazard modeling was used to compare recurrence-free survival between patients who received full-dose induction BCG and those who received reduced-dose induction BCG. All data were stored in secured Excel version 16 for Mac (Microsoft; Seattle, WA, USA). Statistical analysis was performed using R statistical software version 4.1.0 (R Foundation; Vienna, Austria). Statistical significance was defined as *p* < 0.05.

## 3. Results

### 3.1. Patient Cohort and Baseline Clinical Information

A total of 254 patients who received intravesical therapy at our institution were identified and reviewed for this study. One hundred and fifteen patients did not meet the inclusion criteria. Thus, our final cohort was 139 patients (Table 1). The median age was 70 years, and 22% of the total cohort was female. Roughly a quarter (23%) of the cohort was from a minority background (77% white, 17% black, 6% other ethnicity). Fifty-six patients received reduced-dose induction BCG, and the remaining 83 received full-dose induction BCG per AUA guidelines (50 mg BCG delivered weekly for six weeks). Of the reduced-dose cohort, 22 received induction BCG at 1/3 dose (100 mg total), 19 received induction BCG at 1/2 dose (150 mg total), and 16 received a reduced induction BCG dose consisting of 6 doses of either 1/2 or 1/3 dose (median total dose 132 mg [interquartile range: 123–141 mg]).

Patients receiving reduced-dose induction BCG were propensity matched 1:1 to patients receiving full-dose induction BCG therapy. One hundred twelve of 139 (80.6%) patients in the total cohort were newly diagnosed NMIBCa patients, with no differences in rates of prior bladder cancer between the full-dose induction BCG cohort and the reduced-dose induction BCG cohort before and after propensity matching (Table 1). In terms of pathology among patients who received reduced-dose induction BCG, 17/56 (30.3%) were Ta, 8/56 (14.3%) were CIS, 26/56 (46.4%) were T1, and 5/56 (8.9%) had mixed pathology. Before propensity matching, the pathology of patients receiving full-dose therapy was 37/83 (44.5%) Ta, 14/83 (16.8%) CIS, 28/83 (33.7%) T1, and 4/83 (4.8%) mixed pathology, which was similar to the pathology distribution of the reduced-dose induction BCG cohort (*p* = 0.25). After propensity matching, the pathology of patients receiving full-dose therapy was 16/56 (28.6%) Ta, 9/56 (16.1%) CIS, 27/56 (48.2%) T1, and 4/56 (7.1%) mixed pathology, which was similar to the reduced-dose induction BCG cohort with more similar rates of Ta and T1 disease than prior to propensity matching (*p* = 0.97).

### 3.2. Bladder Tumor One-Year Recurrence-Free Survival

Recurrence-free survival was significantly decreased for patients receiving reduced-dose induction BCG compared to those who received full-dose induction BCG, with 38.6% of reduced-dose induction BCG patients developing a bladder cancer recurrence within one year versus 33.7% of full-dose induction BCG patients (*p* = 0.03, Figure 2). When analyzing patients in whom this was their initial tumor, 34% of patients receiving reduced-dose induction BCG developed a recurrence within the one-year follow-up compared to 21% of propensity-matched patients who received full-dose induction BCG therapy (*p* > 0.05). In addition, patients receiving reduced-dose induction BCG had significantly earlier recurrences (median time to recurrence 130.4 days in reduced-dose induction BCG patients vs. median time to recurrence 192.6 days in full-dose induction BCG patients, *p* = 0.04).

## 4. Discussion

This study reports our experience with a “real world” cohort of patients with AUA intermediate- and high-risk NMIBCa treated with full-dose induction BCG versus reduced-dose induction BCG with dosage based on allotment availability and measured recurrences at one year. It was found that patients receiving reduced-dose induction BCG had significantly decreased recurrence-free survival and shorter times to recurrence compared to patients receiving full-dose induction BCG who were propensity-matched for age, sex, pathology and initial vs. recurrent NMIBCa.

There is a need to best utilize the increasingly limited BCG supply. Potential options include decreasing instillation frequency and decreasing the dose of each BCG treatment, although the current evidence for these strategies is mixed. The CUETO group performed a randomized clinical trial of 500 patients with Ta, T1 or CIS NMIBCa receiving two consecutive induction courses (12 doses) of full dose or 1/3 dose BCG and demonstrated a similar mean recurrence percentage between the two groups [22]. While they did not demonstrate a difference in recurrence or progression between the two groups, they did note that patients with multifocal tumors receiving full-dose BCG had fewer recurrences, and among patients with high-risk tumors treated with full-dose BCG, there was a trend toward decreased recurrence rates [22]. Ojea et al. reported another CUETO randomized trial of 430 patients with intermediate-risk NMIBCa treated with low-dose BCG (1/3 standard dose), very low-dose BCG, or mitomycin C that found low-dose BCG to be superior to very low-dose BCG or mitomycin for recurrence [27]. Another study, EORTC-GU 30962, was a randomized controlled trial where 1355 patients with intermediate- or high-risk NMIBCa were studied in two sets of hypotheses: the first predicting reduced efficacy of 1/3 reduced-dose induction BCG as compared to full-dose induction BCG; the second being one or 3 years of maintenance BCG [19]. Overall, they could not reject the null hypothesis of a 10% decrease in the 5-year disease-free rate for 1/3 dose BCG or 1 year of maintenance. Sub-analysis of the 1/3 dose and full dose induction + 1 year of maintenance groups, which more closely fits our cohort, found similar five-year disease-free rates (54.5% 1/3 dose vs 58.8% full dose) [19]. In addition to these randomized trials, Lobo et al. reported a “real-world” retrospective cohort of 563 patients who received either full-dose BCG or 1/3 dose BCG as induction BCG plus two additional instillations within a 6-month period with a median clinical follow-up interval of 54.8 months [49]. In their study, they demonstrated no differences in time to recurrence, time to progression, or cancer-specific survival between patients receiving full-dose BCG and those receiving 1/3 dose BCG, although the cohort was not propensity matched and patients receiving 1/3 dose BCG were more likely to have low-grade disease, solitary tumors, or lack associated CIS compared to patients treated with full-dose BCG, limiting generalizable conclusions as to the real-world efficacy of reduced-dose BCG. By contrast, NIMBUS was a randomized phase III clinical trial investigating a reduced-dose regimen of fewer administrations—3-week induction course (weeks 1, 2 and 6) followed by 2-week maintenance BCG at 3, 6 and 12 months—in patients with HGTa or T1 disease [30]. The authors demonstrated significantly inferior times to first recurrence among those receiving the reduced-dose regimen, resulting in premature study termination.

Arguments have been put forward to explain this discrepancy in results between studies comparing full-dose and reduced-dose BCG regimens. Kamat et al. recently reported a wide variability in the number of viable organisms in each BCG vial [50]. These authors argue that in two hypothetical patients treated with reduced-dose BCG and full-dose BCG from the same manufacturer, the variability of actually delivered BCG is so high that the individual who receives the reduced-dose BCG may theoretically receive more viable BCG organisms than the individual treated with a full-dose BCG vial. Despite this theory of potential dosing equivalence, two recent meta-analyses have reported mixed results. Zeng et al., analyzing 6 randomized controlled and 2 quasi-randomized controlled trials comparing reduced-dose BCG and full-dose BCG, reported no significant difference in time to recurrence between dosing regimens (HR = 1.15, 95%CI: 1.00–1.31, *p* = 0.05) and, on subgroup analysis, no difference in time to recurrence between patients receiving reduced-dose BCG and full-dose BCG as induction therapy alone (HR = 1.15, 95% CI: 0.65–2.04) [28]. In addition, they reported no difference in the time to progression [28]. In analyzing nine randomized controlled trials that compared reduced-dose induction BCG and full-dose induction BCG, Choi et al. reported contrasting results of a significantly higher tumor recurrence rate among patients receiving reduced-dose induction BCG (HR = 1.45, 95%CI: 1.09–1.94, *p* = 0.01) but no differences in tumor progression, cancer-specific survival, or overall survival [29]. However, the authors note that the two studies driving the recurrence difference occurred prior to 2000, with subsequent studies suggesting no recurrence rate difference.

Our study reports pragmatic, short-term, “real-world” data of adapting to a limited BCG supply by treating intermediate- and high-risk NMIBCa with reduced-dose induction BCG instilled according to the classic administration schedule among a propensity-matched cohort. Our results suggest potentially significantly decreased efficacy in terms of one-year recurrence rates compared to full-dose BCG induction. While lack of BCG maintenance or a standardized universal alternative intravesical maintenance regimen is a limitation of this study, Figure 1 illustrates how most recurrences occurred within 100 days of induction therapy prior to the timing of the first treatment in standard intravesical maintenance therapy scheduling, suggesting that the observed recurrence rate difference is potentially due to induction BCG dosage differences. Compared to EORTC-GU 30962, our cohort was not able to receive maintenance BCG therapy, and a higher proportion had primary tumors (87% vs. 61%) and/or T1 disease (45% vs. 36%), factors that likely explain our higher one-year recurrence rate (29% vs. 6.2%) [19]. In addition, many of the older studies, including those analyzed by the meta-analyses, reported pathology in the World Health Organization (WHO) system of grades 1–3 and had almost exclusively European populations, whereas we report staging in Ta/CIS/T1 and had a more diverse, American cohort. The short-term results from our experience using reduced-dose BCG induction therapy for intermediate- and high-risk NMIBCa in a university clinical practice suggest that reduced-dose induction BCG may not be a harmless alternative to full-dose induction BCG and provide urologists with realistic results from one approach of adapting to real-world supply constraints to help guide difficult BCG allocation decisions.

Our institution’s BCG allocation strategy closely resembled AUA/BCAN’s 2020 BCG recommendations and our results, despite being from a relatively small, single-center retrospective study, provide a “real-world” glimpse into the potential outcomes of these recommendations. Importantly, our results demonstrate a worrying trend of early recurrences, most within 100 days of induction therapy and identified on the first post-induction surveillance cystoscopy, among patients receiving reduced-dose induction BCG. Combined with a large body of literature demonstrating inconsistent results with reduced-dose BCG, our study suggests that reducing the dose of induction BCG may cause harm and highlights the need for further research into alternative intravesical therapies. In addition, there is a subset of patients who are BCG unresponsive, with prior studies indicating multifocality, lymphovascular invasion, and high-grade histology on re-TURBT as predictors of poor BCG response for whom effective alternative therapies are needed [51,52,53]. Combination therapy with sequential gemcitabine and docetaxel is a potential alternative. One retrospective report of using this dual regimen as a rescue therapy following recurrence after prior BCG treatment demonstrated 60% one-year recurrence-free survival and 46% two-year recurrence-free survival [40]. In addition, one small early cohort of 18 BCG-naïve high-grade NMIBCa patients treated with sequential intravesical gemcitabine and docetaxel reported a 78% six-month recurrence-free survival. More recently, a single-center retrospective study comparing sequential intravesical gemcitabine and docetaxel versus intravesical BCG as first-line adjuvant therapy in patients with high-risk NMIBCa reported better high-grade recurrence-free survival (hazard ratio, 0.57; 95% CI, 0.33–0.97; *p* = 0.04) and recurrence-free survival (hazard ratio, 0.56; 95% CI, 0.34–0.92; *p* = 0.02) associated with intravesical gemcitabine and docetaxel on multivariate Cox regression analysis as well as a higher rate of treatment discontinuation during induction therapy associated with BCG (9.2% vs. 2.9%; *p* = 0.02) [42]. While potentially promising, this evidence for dual intravesical gemcitabine and docetaxel is early and limited, and more, higher-quality data with longer follow-up is greatly needed that will hopefully be provided in the currently enrolling randomized controlled phase III BRIDGE trial in the coming years [44].

Intravenous pembrolizumab is another potential option for adjuvant therapy for NMIBCa that is currently being investigated, although current results are limited to 101 patients with CIS (37% had concurrent Ta or T1 disease), where 53% of patients demonstrated a complete response duration of greater than 12 months [54,55]. Up-front radical cystectomy with urinary diversion offers a more definitive alternative providing optimal oncologic control but is associated with significant morbidity that might be assumed that many to most patients would likely find objectionable [56,57]. However, the BRAVO feasibility study randomized BCG-naïve patients with high-risk NMIBCa to intravesical BCG or radical cystectomy and reported that quality of life metrics, despite a relative decline at 3 months post-operatively among radical cystectomy patients, were similar between groups at 12 months, although the authors concluded that it would be difficult to recruit patients for a larger trial [58]. Each of these options represents a potential alternative to the BCG dose reduction strategy that has been fraught with inconsistent and potentially worrying data, although more research into these and other potential BCG alternatives is desperately needed to determine how to best treat NMIBCa patients when BCG is limited or unavailable.

Potential limitations of our study include the retrospective, single-center, non-randomized design and relatively small cohort size. There is potential that BCG unresponsive patients were not evenly distributed between the groups given the retrospective nature of the study, although including tumor pathology, which has been shown to be an independent predictor of BCG unresponsiveness, in the propensity-matching procedure mitigated this potential source of variability and the full-dose and reduced-dose groups had similar distributions of pathology [51]. Comparisons to some prior randomized trials are limited by differences in pathologic descriptions and the unavailability of tumor size in our cohort. In addition, due to wide variability in the number of BCG colony forming units in each vial, each patient’s true dose is not known. Our non-standard reduced dose of BCG introduces variability among the reduced dose cohort, although this may more accurately reflect real-world practices where BCG is split by availability at the time of administration. We utilized propensity score matching to help address this source of bias.

## 5. Conclusions

In the setting of an ongoing BCG shortage, urologists have in general been unable to provide full-dose induction and maintenance BCG to all patients in whom it is indicated. Reduced-dose BCG induction therapy for NMIBCa seems to increase the risk of recurrence at one year of NMIBCa compared to full-dose induction therapy when maintenance BCG is not offered. These data provide practical information for the application of reduced-dose BCG based on similar allocation principles specified in the 2020 AUA/BCAN BCG supply shortage recommendations and indicate the possible decreased efficacy of reduced-dose induction BCG for the treatment of intermediate- or high-risk NMIBCa in the short-term. There is a need for further investigation to determine the optimal therapy with current available agents for NMIBCa to address the limited BCG supply.

## Figures and Tables

**Figure 1 cancers-15-03746-f001:**
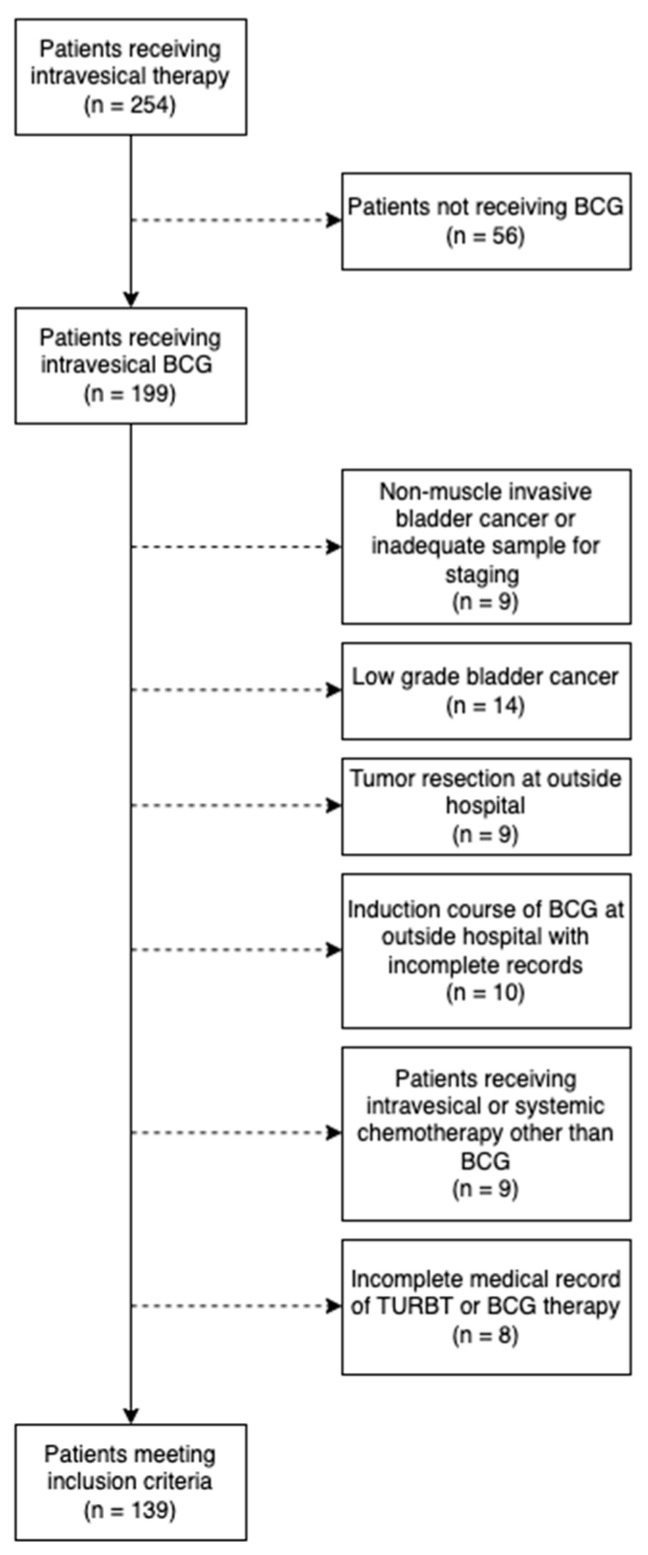
Flow chart of inclusion and exclusion criteria for patients who received induction intravesical BCG at either full or reduced doses.

**Figure 2 cancers-15-03746-f002:**
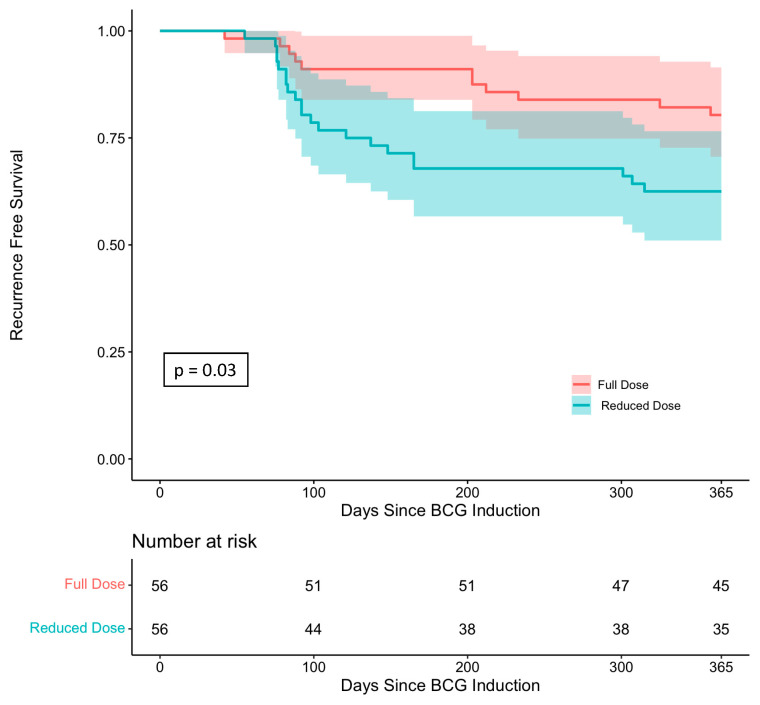
Recurrence free survival Kaplan-Meier curve comparing any reduced-dose vs full-dose induction BCG therapy.

**Table 1 cancers-15-03746-t001:** Patient and pathology characteristics of NMIBCa patients who received full or reduced dose induction BCG before and after propensity score matching.

		Before Matching			After Matching		
		Full Dose (n = 83)	Reduced Dose (n = 56)	*p*	Full Dose (n = 56)	Reduced Dose (n = 56)	*p*
Age (years), median		69.9	70.8	0.60	71.1	70.8	0.61
Sex, n male (% male)		64 (77.1%)	45 (80.4)	0.91	46 (82.1%)	45 (80.4%)	1.0
Ethnicity	White, n (%)	66 (79.5%)	41 (73.2%)	0.67	46 (82.1%)	41 (73.2%)	0.52
	Black, n (%)	13 (15.7%)	11 (19.6%)		7 (12.5%)	11 (19.6%)	
	Other, n (%)	4 (4.8%)	4 (7.1%)		3 (5.4%)	4 (7.1%)	
Prior Bladder Cancer, n (%)		21 (25.3%)	6 (10.7%)	0.05	6 (10.7%)	6 (10.7%)	1.0
Initial Pathology (%)	Ta, n (%)	36 (43.4%)	17 (30.3%)	0.28	17 (30.3%)	17 (30.4%)	0.95
	CIS, n (%)	14 (16.9%)	8 (14.2%)		10 (17.8%)	8 (14.2%)	
	T1, n (%)	28 (33.7%)	26 (46.4%)		25 (44.6%)	26 (46.4%)	
	Mixed, n (%)	4 (4.8%)	5 (8.9%)		4 (7.1%)	5 (8.9%)	

## Data Availability

The datasets generated for this study are available on request to the corresponding author.

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
