# Peer review of "Diminished Short-Term Efficacy of Reduced-Dose Induction BCG in the Treatment of Non-Muscle Invasive Bladder Cancer"

_cancers, 2023, doi:10.3390/cancers15143746_

Round 1

Reviewer 1 Report

The authors retrospectively assessed the recurrence rate of high-risk NMIBC between patients undergone to full dose or reduced dose of BCG. To note, BCG-induction was only performed, and the true reduced dose was not homogeneous in the comparative harm.

In line 208, authors stated that recurrence was defined as new bladder tumour after BCG induction. Please, clarify that primary end-point was recurrence at 1 year like it was computed.

Likewise, In line 205 “alternative maintenance treatments were offered to patients”, clarify if those patients were excluded from analysis.

Finally, a table reporting  patients features is advisable.

non

Reviewer 2 Report

General comment

The manuscript entitled “Diminished short-term efficacy of reduced-dose induction BCG in the treatment of non-muscle invasive bladder cancer” aims to evaluate the efficacy of reduced dose induction BCG compared to full dose BCG regarding recurrence rates. The manuscript deals with an interesting and current topic, considering the recent difficulties related to BCG shortages. Despite being a retrospective study, and therefore having different limitations, it could enrich the literature on the topic. Few corrections are required to improve the overall quality of the manuscript.

INTRODUCTION

55: the issue is related to the grade of bladder cancer. Higher recurrence rates are associated with high-risk BC. Report this difference.

103-140: this part could be moved to the discussion

142-157: redundant, revise.

MATERIALS AND METHODS

173: add IRB number.

200: this is a limitation that should be discussed and reported.

229: how did you assess the normal distribution of data?

244: please specify

DISCUSSION

302-310: avoid redundancies

Briefly add a section regarding the possibility of unresponsive patients to BCG. To this regard please see: DOI: 10.1016/j.urolonc.2022.05.016, DOI: 10.23736/S2724-6051.22.04953-9 and DOI: 10.1016/j.euo.2021.11.006

minor typos

Reviewer 3 Report

Estimated Authors,

I've read with great interest your original article based on the EMR from your parent Healthcare provider (i.e. University of Pennsylvania). In this study, through a small but well matched sample (through propensity score matching strategy) Authors were able to address how a reduced dosage BCG strategy may be quite ineffective in achieving the therapy targets.

In fact, the article is both interesting and well designed, therefore I've only a couple of suggestions rather than requests for its improvement, and more precisely:

1) please make consistent the reporting of decimal figures across tables and main text; p value ideally with 3 figures, OR/HR etc with 2 figures, etc.

2) please edit figure 2 by implementing the number of surviving participants by stage of follow up (see this article for what I'm pointing it out: https://www.google.com/url?sa=i&url=https%3A%2F%2Fwww.nejm.org%2Fdoi%2F10.1056%2FNEJMoa2001342&psig=AOvVaw3ujaQn1iTS1beGOCUsUtri&ust=1687897099544000&source=images&cd=vfe&ved=0CBEQjRxqFwoTCPi9u-rg4f8CFQAAAAAdAAAAABAE)

3) across the article, a sentence alike "because of the ongoing shortage of BCG"; clearly this is the ongoing real world status, but please provide some synonyms. 

No Grammar Issues. Please avoid repetition in some sentences in introduction and discussion about the shortage of BCG.

Round 2

Reviewer 1 Report

the new version of your study has been considerably improved. 

Reviewer 2 Report

Authors improved the manuscript accordingly to previous suggestions. No further issues are reported